# Deep Learning-Based Approach for Autonomous Vehicle Localization: Application and Experimental Analysis

Norbert Markó, Ernő Horváth *, István Szalay and Krisztián Enisz

Vehicle Industry Research Center, Széchenyi István University, H-9026 Győr, Hungary; marko.norbert@ga.sze.hu (N.M.); szalay.istvan@ga.sze.hu (I.S.); enisz.krisztian@ga.sze.hu (K.E.)
* Correspondence: herno@ga.sze.hu

**Abstract:** In a vehicle, wheel speed sensors and inertial measurement units (IMUs) are present onboard, and their raw data can be used for localization estimation. Both wheel sensors and IMUs encounter challenges such as bias and measurement noise, which accumulate as errors over time. Even a slight inaccuracy or minor error can render the localization system unreliable and unusable in a matter of seconds. Traditional algorithms, such as the extended Kalman filter (EKF), have been applied for a long time in non-linear systems. These systems have white noise in both the system and in the estimation model. These approaches require deep knowledge of the non-linear noise characteristics of the sensors. On the other hand, as a subset of artificial intelligence (AI), neural network-based (NN) algorithms do not necessarily have these strict requirements. The current paper proposes an AI-based long short-term memory (LSTM) localization approach and evaluates its performance against the ground truth.

**Keywords:** autonomous vehicle; open source; localization; GNSS; IMU; odometry; self-driving





## 1. Introduction

A crucial aspect of contemporary autonomous vehicle research involves enhancing localization methods [1–4]. Errors accumulate over time during position estimation: even a small error can rapidly ruin the estimation quality within a fraction of a second. Traditional methods like the extended Kalman filter (EKF), unscented Kalman filter (UKF) and particle filter (PF) have served as stalwart approaches in handling non-linear systems, affected by noise in both the system and estimation models. However, these approaches necessitate an in-depth understanding of the non-linear model noise characteristics of the sensors. To properly tune these kinds of systems, a substantial investment of time and effort is needed. For example, to tune various Kalman filter model parameters, e.g., process noise covariance, pre-whitening filter models for non-white noise, and frequently even optimization methods are applied [5–8]. In contrast, neural network-based algorithms offer a promising alternative that circumvents these stringent requirements. It is important to distinguish between two neural networks (NN) involved approaches. The first one uses neural networks as an optimization approach, and only uses the network to obtain the proper parameters for the Kalman filter. The second one perceives the problem as a black box without a pre-existent filter. This paper uses the second approach and introduces a neural network-based localization method, presenting an evaluation of its performance and discusses the theoretical background of the well-established EKF estimation methods. The study emphasizes practical applications and highlights the associated limitations of these methodologies.

### 1.1. Robot and Vehicle Position Estimation

A significant challenge both in robotics and autonomous vehicle research is to determine the robot's current position [9–11] often referred to as localization. Robots use

their sensor data to extract the position information. The sensor data may include speed, acceleration, image, depth information and distance measurements, with different levels of accuracy. By combining information from diverse sensors, a comprehensive position estimation could be achieved. Generally, the accuracy of the robot's position estimation improves with the increasing number of available sensors, especially when the raw sensor data have a certain level of accuracy. In robotics one of the most widely used localization techniques is simultaneous localization and mapping (SLAM) [12]. SLAM is a sensor fusion approach and as the name suggests it provides not only the vehicle's location but also an estimate of the environment. In recent years this approach has become much more robust in exchange for efficiency [12]. SLAM usually involves camera or LiDAR (light detection and ranging) data processing, which means a large amount of sensor data. On the other hand, more efficient and sometimes less robust solutions [2,3,13] only rely on simple measurements such as velocity, acceleration or magnetic field. The most used estimators in this field are EKF, UKF and PF [14]. In the [15] an invariant extended Kalman filter (IEKF) based, loosely coupled IMU-GNSS integration scheme was introduced. The IEKF as a navigation filter using quaternion rotation for the INS-GNSS integration scheme. The observable states of the Kalman filter are made to converge within an area of attraction. There are solutions which fuse IMU and LIDAR data for positioning, while there are solutions that use only LIDAR data for estimation. What both approaches have in common is that LIDAR provides a much larger amount of data than conventional in-vehicle sensor systems. As a result, their computational requirements are also more significant. This approach makes data processing and data optimization one of the main challenges, and the ability to estimate as accurately as possible from the reduced data volume. Such odometry estimation solution was presented in [16] which approach relies on point-to-point ICP (iterative closest point) combined with adaptive thresholding.

Compared to the previously introduced localization, a much broader problem set is called system state estimation. This is a fundamental concept in dynamic systems, which helps to understand and predict the behavior of complex, dynamic processes. Dynamic systems are prevalent in numerous fields, spanning from engineering and physics to economics and environmental sciences, and often involve evolving states influenced by various inputs, disturbances and inherent uncertainties. The primary objective of state estimation is to ascertain the unobservable or partially observable states of a dynamic system based on available measurements and a mathematical model that describes the system. Some prevalent approaches for this problem set involve probabilistic and statistical methods such as the Bayes filter, Kalman filter [13], (extended and unscented variations), Gaussian approaches [17].

The paper [18] suggests enhancing the Kalman filter by incorporating the distance attributes of both LIDAR and radar sensors. An analysis of the sensor properties of LIDAR and radar with respect to distance was conducted, leading to the development of a reliability function to expand the Kalman filter's capability in capturing distance-related features. In this study real vehicles were used as an experiment, and comparative measurements were performed that integrated sensor fusion with a fuzzy, adaptive noise measurement and the Kalman filter. The experimental findings demonstrated that the approach employed in the study produced reliable position estimates. Similarly, in [19] a localization approach that integrates an inertial dead reckoning model with 3D LIDAR-based map matching is suggested and empirically validated in a dynamic urban setting encompassing diverse environmental conditions.

In the study presented in [20], the unique challenges faced by UAVs in outdoor wilderness scenarios are addressed with a significant focus on enhancing received signal strength (RSS) for accurate positioning and collision avoidance. The presence of colored noise in RSS data disrupts traditional least squares (LS) algorithms. The extended Kalman filter (EKF) emerges as a solution, wherein a colored noise model is ingeniously incorporated, sharpening its distance estimation accuracy. Alongside, an adaptive algorithm is introduced to account for fluctuating path-loss factors during UAV flights. This comprehensive

EKF-based approach not only ensures precise UAV positioning but also incorporates an orthogonal rule for advanced collision detection and trajectory modification. Experimental findings validate the robustness of this EKF-centric methodology, endorsing its value for collaborative UAV operations.

Increasing the positioning accuracy can be based not only on internal sensor and LIDAR data, but also on increasingly complex and widespread vehicle-to-vehicle or vehicle-to-infrastructure communication networks. If a vehicle within a network loses its GNSS signal, it can obtain positioning data from other nodes in the network. As an example, methods can be based on Vehicular ad hoc networks (VANETs). VANETs communicate additional position data and attempt to achieve localization using the received signal strength indicator (RSSI) analytical model [21,22].

The disadvantage of these methods is that RSSI is significantly affected by the weather and objects in the environment. Several correction methods have been developed to deal with this, including weighted centroid localization (WCL) and signal to interference noise ratio (SINR) methods [23].

There are procedures, similar to the fusion of IMU and GNSS data, which are based on Kalman filter and use RSSI, SINR calculations together with WCL procedure [14].

*1.2. Knowledge-Based Versus Black-Box Modeling*

The mentioned knowledge-based solutions (most notably the EKF) have obtained a good reputation in signal filtering. The main drawback of knowledge-based methods is their dependency on the underlying model accuracy. Kalman filters typically use kinematic models in lower velocities, but in higher velocities, dynamic models are a better fit. These have several parameters that depend on external factors and may vary over the life of the vehicle, for example vehicle mass, weight distribution and tire stiffness. The changes can be neglected, but still the noise characteristics of the model and the sensors must be known and considered in the modelling. A further difficulty is that there are sensor signals, such as GNSS data, where the noise characteristics are not deterministic and can change very quickly. For example, when communication is suddenly lost with the remote base-towers required for real-time kinematics (RTK) based solutions [24], the accuracy and hence the noise characteristics of the GNSS signals can change significantly within seconds. This also shows that with the number of measured data and internal state variables, not only does the computational complexity is increasing significantly due to the significant matrix operation running in the background, but also the complexity of the models makes them difficult to handle. A further problem is that while data from IMU and odometry data can be taken into account in a dynamical or kinematic model directly or through physical relationships, this is more difficult to do for a camera signal or LIDAR point cloud.

In recent years, black-box modelling-based technologies have gained significant traction too. In the field of vehicle localization a numerous papers [24–31] have been published. Maybe even more attention is on smartphone-based pedestrian tracking [32–36]. In this field similar technologies can be applied, but the quality and type of usable sensor set is divergent. From the mentioned approaches [31] is the closest to ours. The main difference is that instead of convolutional neural networks (CNN), our solution uses LSTM. The most known advantage is the ability to accurately predict outcomes and behavior without requiring a deep understanding of the underlying internal mechanisms or complexities. One of the main drawbacks of black-box methods is to provide the right results in all situations. Moreover, they require a large amount of measurement data to teach them, and these measurements must be recorded under the most varied conditions possible.

## 2. Problem Statement and Motivation

In simple terms our aim is to determine the local pose (position and orientation) of the vehicle without GNSS. This also means that during motion, measured signals such as velocity of the vehicle, steering angle, angular velocities and magnetic field vector are present. Velocity and steering angle are provided by the vehicle's CAN bus and the other

signals are from an inertial measurement unit (IMU). The GNSS serves as the ground truth pose, commencing from zero. In this case the GNSS is enhanced with a real-time kinematics (RTK) technique [37]. This technique enhances the precision and accuracy of position data, by utilizing a fixed base station and internet connection. To highlight the nature of the problem, the motion model of the vehicle will be described.

The goal is to determine the vehicle position ($x$ and $y$ coordinates) and heading (yaw angle $\psi$) in two dimensions. This can be performed by fusing IMU measurements with dead reckoning, based on vehicle sensors. To be more precise, in our case odometry is used, which is a specific method of dead reckoning applied in robotics and vehicle navigation. It estimates the vehicle's or robot's position and orientation by analyzing data from motion sensors, typically wheel speed sensors or IMUs. The results will be in local coordinates, commencing from zero $x$, $y$ and $\psi$. To introduce the problem, Figure 1 shows an overview of the physical quantities.

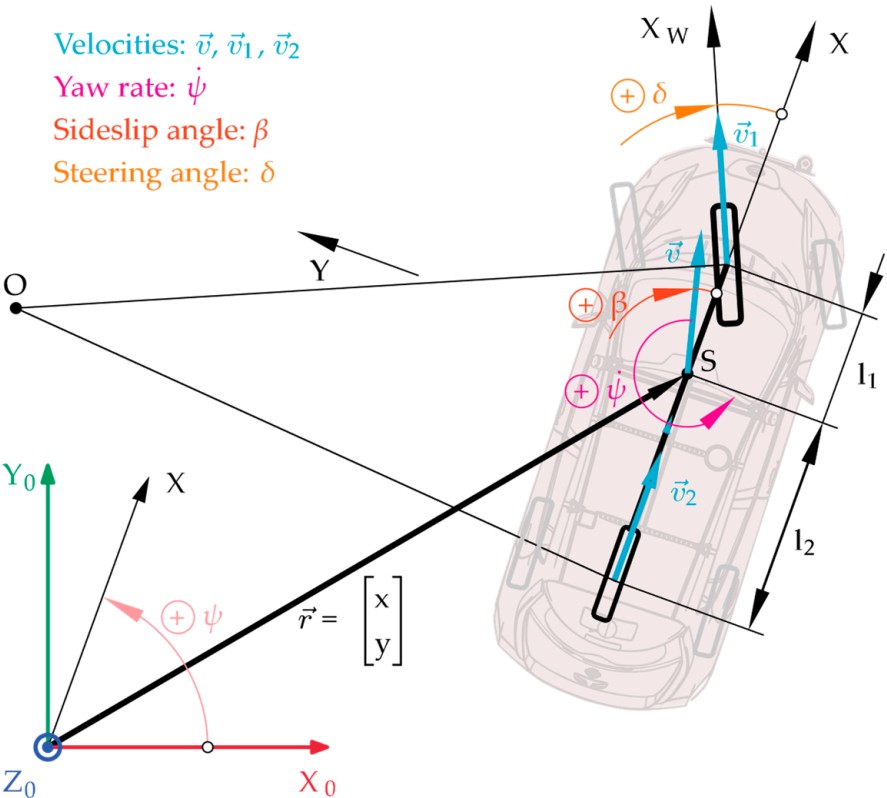

**Figure 1.** Illustration of the single-track kinematic vehicle model. The plus sign indicates the positive reference direction.

On Figure 1, $S$ is the center-of-gravity and $\vec{v}$ is the vehicle's velocity. The motion is tracked in the $X_0$, $Y_0$ Earth-fixed coordinate system, the position vector consist of the $x$ and $y$ coordinates while the vehicle's orientation is described by the yaw angle $\psi$. The yaw rate $\dot{\psi}$ is the measure of how fast the vehicle is rotating or turning around $S$. $\beta$ is the side-slip angle that measured between the direction of motion and the vehicle's longitudinal axis. In the single-track kinematic model, at any given discrete time $k$, the $\delta_k$ steering angle directly determines the $\beta_k$ side-slip angle according to

$$\beta_k = arctan\left(\frac{l_2}{l_1 + l_2}\tan \delta_k^M\right) \tag{1}$$

The kinematic model can estimate in any given $k$ discrete time $x_k$ and $y_k$ position and the orientation $\psi_k$ as follows:

$$\begin{bmatrix} x_k \\ y_k \\ \psi_k \end{bmatrix} = \begin{bmatrix} x_{k-1} \\ y_{k-1} \\ \psi_{k-1} \end{bmatrix} + \begin{bmatrix} \Delta x_k \\ \Delta y_k \\ \dot{\psi}_k T_S \end{bmatrix} = \begin{bmatrix} x_0 \\ y_0 \\ \psi_0 \end{bmatrix} + \sum_{i=1}^{k} \begin{bmatrix} \Delta x_i \\ \Delta y_i \\ \dot{\psi}_i T_S \end{bmatrix} \tag{2}$$

This model and the RTK GNSS will serve as a basis of the comparison. The goal of the research can be summarized as estimating the most accurate pose possible without using a global source (e.g., GNSS), high-bandwidth sensors (e.g., LIDAR) or knowledge-based solutions (e.g., EKF). The resulting method could be used in our lightweight vehicles, research vehicles or outdoor mobile robot platforms. These diverse robotic and vehicular platforms will benefit the robustness, infrastructure independence and computational efficiency from the resulting method.

### 3. Proposed Solution

The problem was approached from the angle of deep learning. The inertial measurement unit (IMU) is a considerably noisy sensor. In spite of that, it contains large amount of useful information with respect to the trajectory to be estimated. By teaching the network generalized patterns on localization on a large enough dataset, we might be able to omit manually defined dynamic vehicle models.

To learn these localization patterns, we used GNSS position data as labels, which have coordinates calculated from a certain origin. These coordinate values are unfortunately meaningless for the learning algorithm. For example, a trajectory part with the same curve characteristics and the same IMU data measurements can correspond to completely different GNSS labels. Therefore, we standardized the labels. The main goal of the learning algorithm is to learn the relative change between two timesteps for two consecutive sensor time step inputs. This can be extended for several timesteps with the right use of recurrent neural networks.

Our network is explained by looking at the most important aspects for reproducibility. These are the network structure, the pipeline in which the network is utilized, and the learning process through which we use the network pipeline for training the model. The idea behind this network was to reduce the number of network parameters as much as it is possible without accuracy loss. This is advantageous, since a low number of parameters encourage generalization if underfitting is monitored. Moreover, a smaller network is easier to use in a real-time setting because of the lower computational demand, which is one of our future goals.

Our deep learning-based solution (Figure 2) consists of a long short-term memory (LSTM) module and two multi-layer perceptrons (MLP) specifically. These three network parts serve different purposes in the learning process. The core of the network is the LSTM module. This is our recurrent module, which is responsible for learning temporal features. In this case, these features are the changes in input sensor readings that correspond to the relative changes between the labels. To provide the input sensor readings in the right format for the LSTM, an MLP module is used. This transforms the raw sensor input into more meaningful features. The second MLP module's purpose after the recurrent part is twofold. On the one hand, it transforms the recurrent part's output into the right format for learning and inference, in this case, three output neurons. On the other hand, it provides further feature transformations which can help with generalization. To fit the network dimensions together, the MLPs are taking three-dimensional inputs. This means that time steps are also part of the input dimension alongside batch size and feature size. This makes the input three-dimensional also. The output is two dimensional, it takes the changes for our three relevant pose values for every timestep.

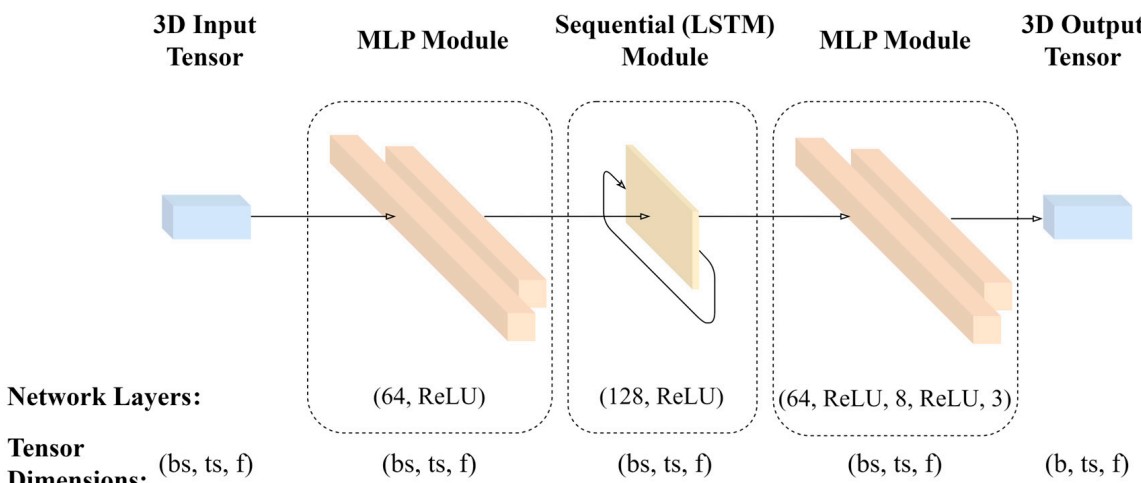

**Figure 2.** Long short-term memory model architecture used in the mentioned approach. The of layer is indicated on the top of the module, in the network layer row, we show the sizes and the activation functions in order. Abbreviations: bs: batch size, ts: timesteps, f: number of features.

There is yet another aspect of recurrent networks that is vital to discuss for this solution called statefulness. Using the network in a stateful way allows it to pass the hidden state of the network to be passed on between batches. If the architecture is not stateful, the state is reset between consecutive batches. We decided to make our network stateless. The reason for this, is that a stateful implementation requires several constraints with regards to training and the sequence of training data. For example, we used a larger batch size of 32 and data shuffling, but this would not have been possible with the stateful variant since it requires ordered data and a batch size of 1 so state can be passed down between sequences.

With regards to training pipeline, it takes inertial, steering and velocity data as input and returns the estimated pose data ($x$, $y$, $\psi$) as output. There is an offline preprocessing phase before the network can be trained or used for inference, where the previously recorded data are mined, filtered and transformed for the relevant task at hand. The learning process itself takes the relative labels in a one-to-many fashion and predicts the next relative change. These relative changes are integrated together when testing is performed. The network LSTM's hidden size was 128, the MLP head layer sizes are 64, 8 and 3. The input (encoding) MLP size was 64. Dropouts are used with a value of 0.5 between MLP layers. The used sequence length was 5. Adam optimizer was employed with a learning rate of $10^{-4}$.

### 3.1. Data Acquisition

To evaluate our approach, it was necessary to conduct real measurements and analyze the experimental results. Our research center has access to the ZalaZone [38] vehicle proving ground. The proving ground facility is located near Zalaegerszeg, Hungary, and one of its main objectives is to support research related to autonomous driving.

The real measurements were carried out with a Nissan Leaf electric vehicle. This test vehicle is equipped with several sensor systems designed to support various autonomous vehicle research activities (Figure 3, Table 1). In this research only GNSS, IMU, wheel velocity and steering wheel orientation were used. The data acquisition subsystem of the vehicle is based on ROS 2 [39]. Robot Operating System, or ROS, is a framework widely used in robotics and autonomous systems development. It is an open-source framework that facilitates the development of autonomous systems through many additional program libraries. The measured data were recorded in MCAP [40] file format, which is the default ROS 2 bag storage format since ROS 2 Iron. This data format can be processed in C++, Go, Python 3, Rust, Swift and TypeScript.

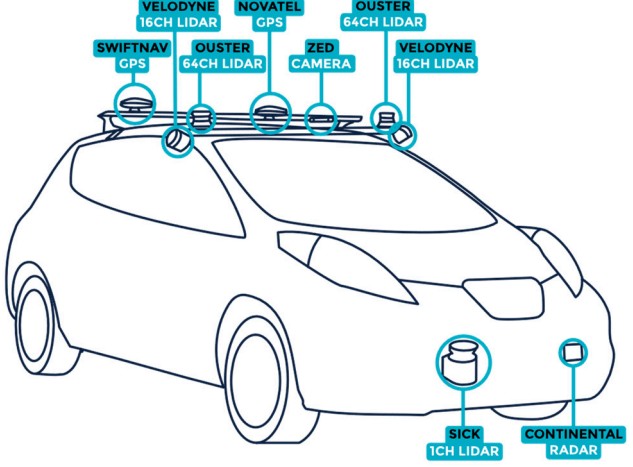

| Sensor | Device |
|---|---|
| 64 channel LIDAR | Ouster OS1-64 |
| 16 channel LIDAR | Velodyne VLP16 |
| 1 channel LIDAR | Sick LMS111 |
| Camera | Stereolabs ZED |
| GPS | SwiftNav Duro Inertial |
| GPS | NovAtel PW7720E1-DDD-RZN-TBE-P1 |

**Figure 3.** The test vehicle with the equipped external sensors.

**Table 1.** The vehicle parameters used during the modelling.

| Parameter | Description | Value |
|---|---|---|
| m | mass with driver and passengers | 1640 kg (with driver and passengers) |
| $l_1$ | CoG distance from the front axle | 1.162 m |
| $l_2$ | CoG distance from the rear axle | 1.538 m |

One of the main drawbacks of LSTM and CNN-based solutions is that a large number of measurements under varying conditions are required in order to properly train the network. However, due to the permanent access to the ZalaZone test track and the own test vehicles, they are preferred to model-based solutions in the current research.

In order to be able to teach the networks properly, a large number of measurements were performed at slower (20 km/h) and faster (90 km/h) speeds. In addition to the ZalaZone University track, which is comparable to a closed test track, the data were recorded on the ZalaZone Smart city test track section, which supports the development of urban autonomous systems (Figure 4).

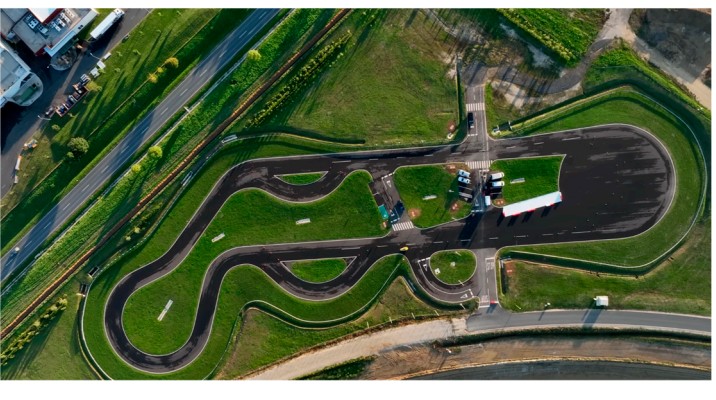

(**a**)

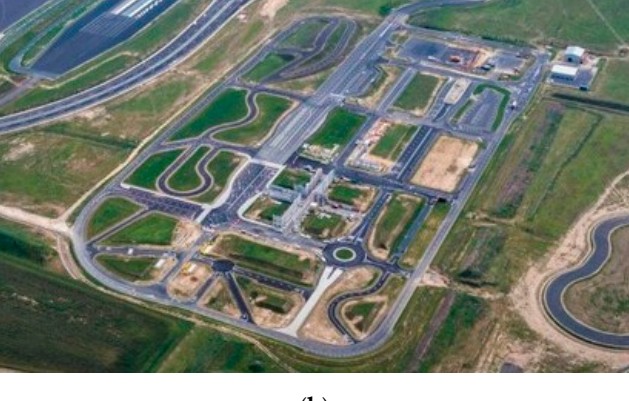

(**b**)

**Figure 4.** Aerial view of the part of ZalaZone proving ground which were used during the measurements: (**a**) ZalaZone University track; (**b**) ZalaZone Smart city test track (taken by the authors).

As the traffic and topography conditions on these sections were not sufficiently varied, data were also collected under real urban conditions within the city of Zalaegerszeg and on the road sections connecting the proving ground to the city (Figure 5).

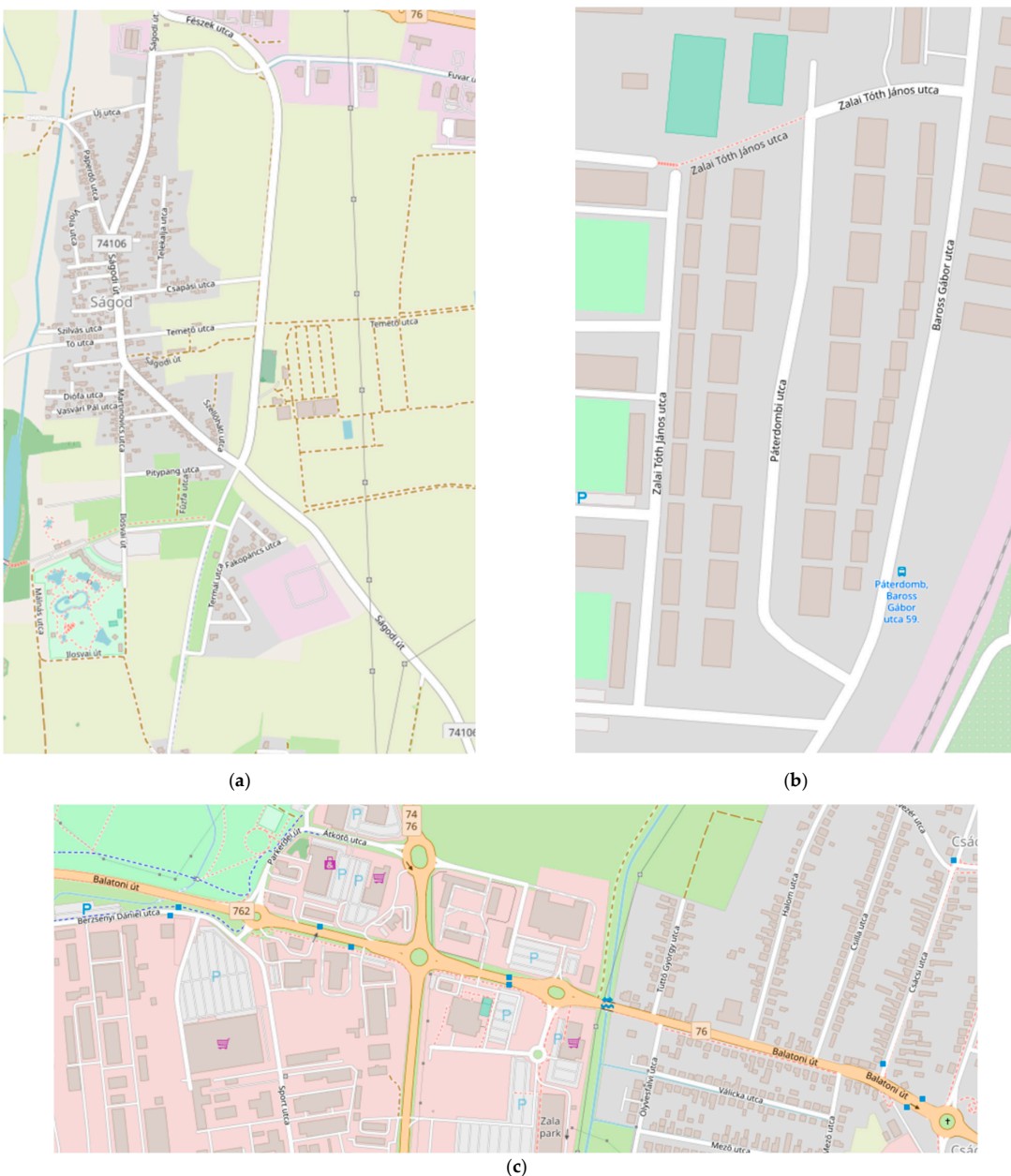

**Figure 5.** Map of Zalaegerszeg where the measurements were performed: (**a**) Ságod; (**b**) Páterdomb; (**c**) Balatoni street. Source: openstreetmap.

It is important to note that the training data (Table A1) used in this research are available to download at jkk-research.github.io/dataset (accessed on 8 December 2023) as written in the Data Availability Statement.

### 3.2. Data Preprocessing

To use acquired the data in the LSTM architecture, they need to be run through a preprocessing script where several transformations happen. The data are upsampled to the sampling frequency of the inertial measurement unit. Other alternatives were downsampling or batching of the higher frequency sensor data, but upsampling preserves the raw training data amount while keeping the ground truth intact as well. We handle the time between sensor reading steps in a discrete way, so training with the upsampled dataset creates a model that works with sensors of this frequency or sensor data upsampled to it. After filtering and upsampling, the kinematic odometry model is calculated for each

of the trajectories and then the ground truth data are transformed to relative labels. The preprocessed training and ground truth data are saved to data frames for easier and faster streaming during training. The Swift Navigation Duro Inertial and NovAtel PW7720E1-DDD-RZN-TBE-P1 RTK based position measurements were saved as ground truth data.

## 4. Evaluation and Results

During training we evaluated our experiments by plotting the predicted and the ground truth position (*x*, *y*) on the same graph, thus obtaining a rough estimation of the prediction accuracy. These plots were also supplemented by the results of the kinematic odometry model for the given trajectory [41]. The prediction components were also plotted individually.

### 4.1. Loss Function

The Odometry Loss function used for our experiments is articulated through two principal terms, each scaled by a distinct weighting factor to confer priority to the more critical term. Initially, the positional loss term ($\mathcal{L}_{position}$),

$$\mathcal{L}_{position} = \frac{1}{N}\sum_{i=1}^{N} \left| p_{pred}^{(i)} - p_{true}^{(i)} \right|^2 \tag{3}$$

employs a Mean Squared Error measure to quantify the discrepancy between the predicted ($p_{pred}$) and true positional vectors ($p_{true}$) respectively, across all *N* instances. Sequentially, the angular error ($\Delta\psi$)

$$\Delta\psi = \psi_{pred} - \psi_{true} \tag{4}$$

is computed as the difference between the predicted ($\psi_{pred}$) and true angles ($\psi_{true}$). This error is then wrapped to the interval $[-\pi, \pi]$ in the following equation:

$$\Delta\psi = \frac{\Delta\psi + \pi}{mod(2\pi)} - \pi \tag{5}$$

to ensure consistency in error measurement across the angular domain. Following this, the angular loss term ($\mathcal{L}_{angle}$)

$$\mathcal{L}_{angle} = \frac{1}{N}\sum_{i=1}^{N} \left(\Delta\psi^{(i)}\right)^2 \tag{6}$$

is evaluated by averaging the squared angular errors across all instances. Finally, the total odometry loss ($\mathcal{L}_{odometry\ loss}$)

$$\mathcal{L}_{odometry\ loss} = w_{position}\mathcal{L}_{position} + w_{angle}\mathcal{L}_{angle} \tag{7}$$

is derived as the weighted sum of the positional and angular loss terms, employing the weighting scalars ($w_{position}$, $w_{angle}$) to balance the contribution of each term towards the overall loss.

### 4.2. Metrics

The metrics used in this paper are the absolute pose error (APE), relative pose error (RPE) and the root mean squared error (RMSE). The absolute position error

$$\text{APE} = \left| p_{true} - p_{pred} \right| \tag{8}$$

straightforwardly measures the absolute discrepancy between the predicted ($p_{true}$) and true positions ($p_{true}$) without normalizing by the true position's magnitude. Subsequently, the relative position error

$$\text{RPE} = \frac{\left| p_{true} - p_{pred} \right|}{\left| p_{true} \right|} \tag{9}$$

quantifies the relative error in the estimated position by computing the ratio of the absolute difference between the predicted and true positions respectively, to the magnitude of the true position. Lastly, the root–mean–square error

$$\text{RMSE} = \sqrt{\frac{1}{N}\sum_{i=1}^{N}\left(p_{true}^{(i)} - p_{pred}^{(i)}\right)^2} \tag{10}$$

provides a measure of the average magnitude of error between the predicted values and the true values for all instances by squaring the differences, averaging them and taking the square root. This error metric is particularly useful in emphasizing larger errors over smaller ones due to the squaring operation.

### 4.3. Experimental Results

The main method of evaluation was visual comparison of the measured and estimated position, aided with the line plots of the individual positional components. The angle component was also visualized with a line plot. All these evaluations were aided with the introduced numerical metrics.

We used sequential processing for the testing data since the lack of shuffling does not affect the weights when they are already tuned and frozen. To visualize the training data, we made sure that every sequence in the shuffled batches is seen at least once, after that, they were collected based on their names and sequences IDs given at data caching.

The results seen on Figures 6 and 7 are the plots of our testing results. One of our testing trajectories is a closed loop and the other one is a more rectangular open path. Figure 6 shows that the loop closure is not perfect, but it is at a very close vicinity compared to the ground truth.

Drift errors and corresponding components can be further inspected in Figure 8. The drift error is increasing on average as we go further and further down the trajectory path, but results suggest that this problem can be solved with a well-trained network.

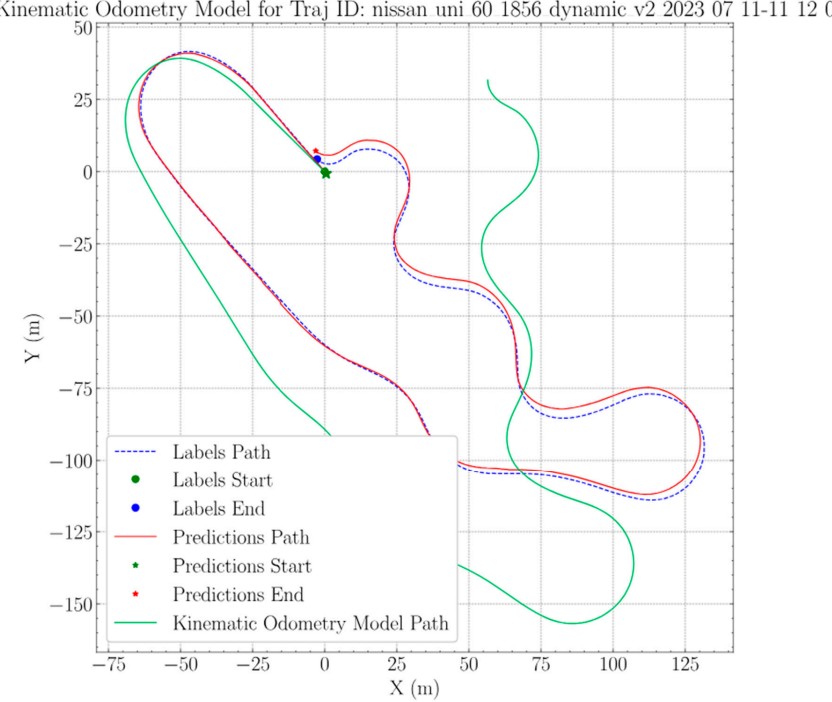

**Figure 6.** Illustration the predicted positions on closed-loop test trajectories alongside the estimated path derived from the kinematic odometry model. The label path is the ground truth data collected by a precise RTK GNSS. The prediction is our proposed method. The odometry is the path generated by a kinematic model (Predictions Start is at the same spot as labels start).

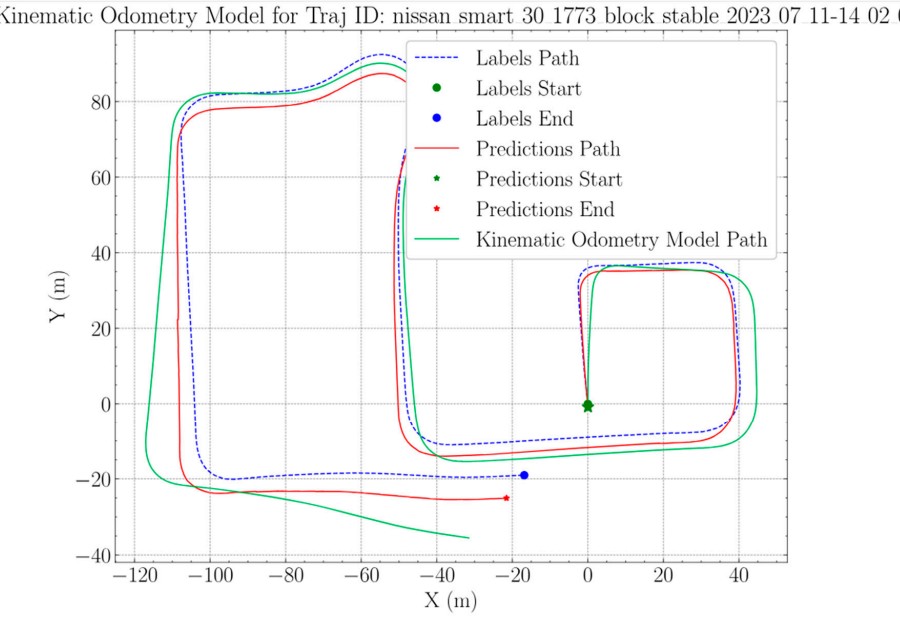

**Figure 7.** Visualized position prediction result on open loop test set trajectories along with the estimated kinematic odometry model path (Predictions Start is at the same spot as labels start).

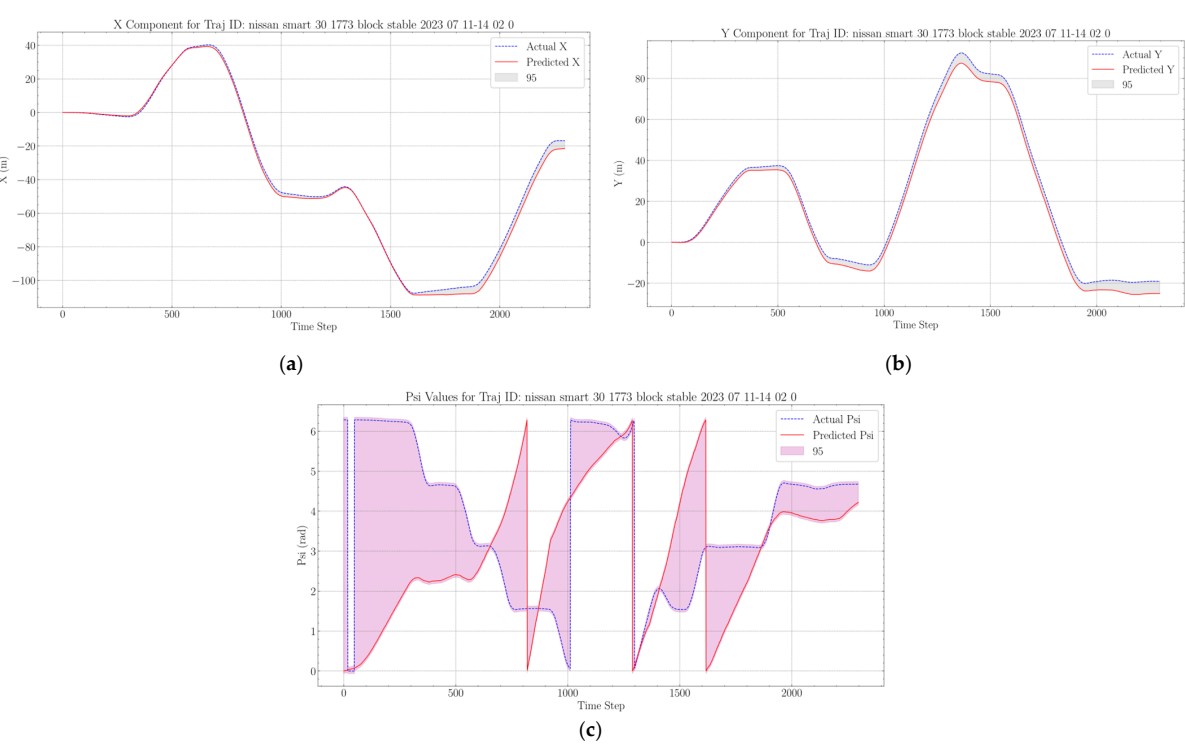

**Figure 8.** Corresponding components of the open loop test set trajectories: (**a**) X component; (**b**) Y component; (**c**) Yaw angle.

One of our goals is to train a network that works well in a real-world environment and these collected experiment results help us build on the knowledge while already presenting a working version.

One of the weaknesses of the trained model is the estimation of the angle component. There are sections where the estimated value changes are on point, but the measured values are going in the opposite direction. This stems from the fact that radian estimation can be sensitive to small changes even with the constraints in place (e.g., it can only change between 0 and $2\pi$).

Figure 9 shows us a full trajectory from the training dataset and corresponding components in Figure 10. These figures were included to present a sample from the theoretically obtainable results.

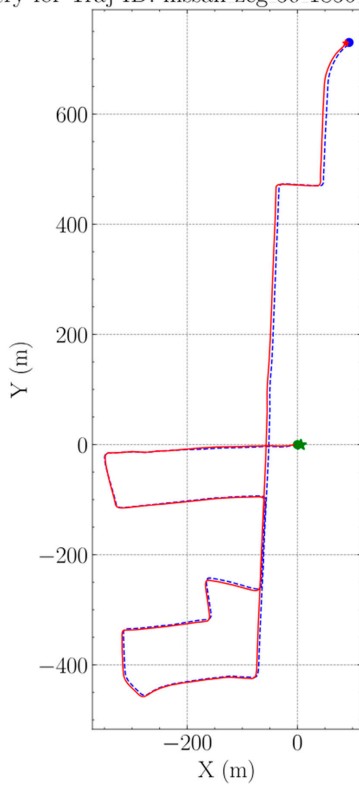

**Figure 9.** Training example for trajectory data. Notations: Green Dot: Labels Start, Blue Dot: Labels End, Green Star: Predictions Start (at the same spot as Labels Start), Red Star: Predictions End.

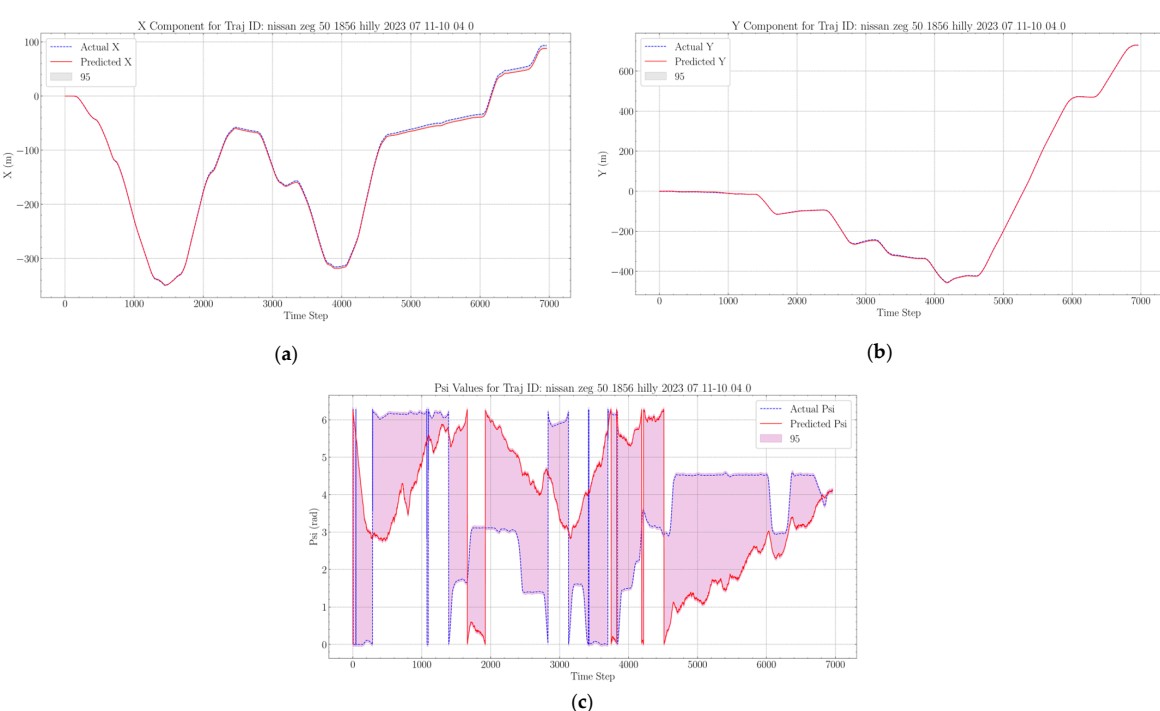

**Figure 10.** Corresponding components of the training example: (**a**) X component; (**b**) Y component; (**c**) Yaw angle.

Results on the training dataset are always better compared to the validation and testing results, since it learns the patterns on this data and there will always be a small amount of overfitting with respect to these samples, even when the generalization is good. Apart from that, it can be seen that the drift error is almost completely gone in these cases, even when we integrate the results, without communication between the individual sequence pieces. This attribute showcases the potential in these types of deep learning-based algorithms. The problems with angle estimation are present here as well, so this is an aspect of the estimation that needs more attention in the future.

The quantitative results presented in Table 2 are based on the metrics described at the start of the section. We divided the numerical evaluation into training and testing values and ran the entire training dataset trough network inference. The values show that the testing is close enough to the training, so combined with the visual results we can say with adequate certainty that the network learned useful general patterns that can be utilized outside of the training data.

**Table 2.** Results based on our chosen evaluation metrics.

| Dataset | RPE (pos) | RPE (angle) | APE (pos) | APE (angle) | Root Mean Squared Error (RMSE) | Loss |
|---------|-----------|-------------|-----------|-------------|--------------------------------|------|
| training | 0.031 | 0.03 | 0.03 | 0.025 | 0.126 | 0.016 |
| test | 0.045 | 0.04 | 0.013 | 0.025 | 0.163 | 0.027 |

## 5. Discussion

One of the main purposes of our work was to avoid Kalman filter tuning when it comes to position estimation and odometry to simplify the testing of integrated algorithms for autonomous vehicles. The proposed deep learning-based approach opens the possibility to simplify position estimation workflow. We have proven that our proposed approach outperforms the Kinematic Odometry Model even on limited training data. Our network predicts the position and angle of a vehicle over long sequences with far greater accuracy compared to the aforementioned Kinematic Odometry Models. Since our solution was proposed as a possible alternative for traditional algorithms such as the EKF, the advantages and the downsides should be discussed.

One of the biggest advantages of our neural network-based approach is the absence of algorithm tuning for testing and integration. This is a long-standing problem with Kalman filter-based approaches. With sufficient training, the neural network model can learn the patterns necessary for robust performance in diverse test scenarios [1,4]. On top of that, by improving the model architecture and increasing its parameter number in tandem, the learning algorithm may be able to identify nuanced patterns that might be hard to include in traditional algorithms. We are planning on investigating and improving on the neural network-based approach further. The field is also promising because of the ever-increasing computational capacity and the improvements in the field of deep learning, both of which might hasten the rate of improvement in neural network-based localization tasks.

The proposed solution's downside lies in its very difference, which is the substantial data requirement. Since deep learning algorithms are not based on physical relationships, it is difficult to predict how they will respond to circumstances not covered in training. Generating a fitting and sufficient training dataset encompassing diverse scenarios often proves to be a challenging task. This means that although the intended goal of eliminating Kalman filter tuning was achieved, another obstacle arises from the training dataset creation. As a rule of thumb, the proposed solution can be a decent option if the measurements necessary for the training can be easily obtained. Otherwise, the classical approaches such as the Kalman filter may be better suited for localization tasks. This is a trade-off between the two approaches.

The employed LSTM architecture proved to be a good candidate for position estimation even with a limited number of parameters. This low parameter count may be

advantageous for our future real-time experiments but could prove to be a disadvantage when it comes to generalization capabilities.

To boost the capabilities of our estimation method, we are planning on implementing a transformer network. Transformer networks solve the problem of slower sequential processing, can store more temporal information and have far better generalization capabilities compared to traditional sequential networks. The only drawback is the slower processing speed, since one of our potential future goals is to enhance the approach's capability to operate in real-time.

Both traditional recurrent networks like LSTMs and GRUs, as well as more recent architectures like transformers, can be effectively used for real-time applications. Consideration here is that transformer architectures are more computationally demanding. This might hinder real-time performance, but increasing computational capacity can be observed even on edge devices, where testing usually happens. Efforts can also be seen from the software side to make calculations in transformer layers more efficient. Both improvements might make the dilemma obsolete in the near future.

Possible future work includes making improvements on the neural network architecture for better generalization capabilities and making the network capable of real-time predictions. Our future goals are data collection on a larger scale for the task and improving the model from the mentioned aspects.

It is also important to mention that the presented approach is sensitive to changes in sensor sampling frequencies. Since the data used as input were upsampled to the frequency of the IMU, we can only provide the data at this frequency for the network to work correctly. This can be solved by not treating time as discrete steps, but by using the time difference in our future calculations.

**Author Contributions:** Conceptualization, N.M., E.H., I.S. and K.E.; methodology, N.M.; software, N.M., E.H. and I.S.; validation, N.M., E.H., I.S. and K.E.; formal analysis, N.M.; investigation, E.H. and N.M.; resources, K.E. and E.H.; data curation, E.H.; writing—original draft preparation, N.M.; writing—review and editing, N.M., E.H., I.S. and K.E.; visualization, N.M. and E.H.; supervision, K.E.; project administration, E.H.; funding acquisition, E.H. All authors have read and agreed to the published version of the manuscript.

**Funding:** The research was supported by the European Union within the framework of the National Laboratory for Artificial Intelligence (RRF-2.3.1-21-2022-00004).

**Data Availability Statement:** The datasets related to our work in the paper are publicly available at: jkk-research.github.io/dataset (accessed on 5 December 2023).

**Conflicts of Interest:** The authors declare no conflict of interest.

## Appendix A

**Table A1.** Parameters and description of the training data files.

| Location | Maximum Velocity | Passengers | Description | Measurement File Name |
|---|---|---|---|---|
| ZalaZone University track | 60 km/h | 3 | Test track: Full length | nissan_uni_60_1856_dynamic_2023_07_11-11_08_0.mcap |
| ZalaZone University track | 60 km/h | 3 | Test track: Full length | nissan_uni_60_1856_dynamic_v2_2023_07_11-11_12_0.mcap |
| ZalaZone University track | 40 km/h | 2 | Test track: Full length, more curves involved | nissan_uni_40_1769_curvy_2023_07_11-11_15_0.mcap |
| ZalaZone University track | 30 km/h | 3 | Tortuous track set | nissan_uni_30_1844_curvy_2023_07_11-11_22_0.mcap |
| ZalaZone University track | 40 km/h | 4 | 3 round test set | nissan_uni_40_1931_repeated_stable_2023_07_11-11_26_0.mcap |
| ZalaZone University track | 50 km/h | 4 | Tortuous, 3 round | nissan_uni_50_1931_repeated_dynamic_2023_07_11-11_31_0.mcap |
| ZalaZone University track | 20 km/h | 3 | Slow, tortuous test set | nissan_uni_20_1848_curvy_2023_07_11-11_37_0.mcap |
| ZalaZone Smart city | 30 km/h | 2 | Tortuous test set | nissan_smart_30_1773_block_stable_2023_07_11-14_02_0.mcap |

**Table A1.** *Cont.*

| Location | Maximum Velocity | Passengers | Description | Measurement File Name |
|---|---|---|---|---|
| ZalaZone Smart city | 50 km/h | 3 | Tortuous, higher speed in the straight periods | nissan_smart_50_1856_diverse_2023_07_11-14_10_0.mcap |
| ZalaZone Smart city | 40 km/h | 2 | Normal speed, tortuous, 2 rounds | nissan_smart_40_1769_blocks_2023_07_11-14_19_0.mcap |
| ZalaZone Smart city | 70 km/h | 4 | Normal speed, tortuous | nissan_smart_70_1931_diverse_dynamic_2023_07_11-14_26_0.mcap |
| ZalaZone Smart city | 80 km/h | 3 | Tortuous, high acceleration at the end | nissan_smart_80_1848_short_straight_sections_2023_07_11-14_32_0.mcap |
| Zalaegerszeg | 50 km/h | 3 | Inside the town, hill section/Ságod | nissan_zeg_50_1856_long_straight_sections_2023_07_11-09_35_0.mcap |
| Zalaegerszeg | 90 km/h | 3 | Between the test track and the town, long and mostly straight | nissan_zeg_90_1856_high_speed_2023_07_11-09_23_0.mcap |
| Zalaegerszeg | 50 km/h | 3 | In the town, Balatoni street | nissan_zeg_50_1856_curvy_flat_2023_07_11-09_54_0.mcap |
| Zalaegerszeg | 50 km/h | 3 | Hill section in the town, Páterdomb | nissan_zeg_50_1856_hilly_2023_07_11-10_04_0.mcap |

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
