# Peer review of "Deep Learning-Based Approach for Autonomous Vehicle Localization: Application and Experimental Analysis"

_machines, doi:10.3390/machines11121079_

Round 1

Reviewer 1 Report

Comments and Suggestions for Authors

The method proposed in the article is informative for the research of autonomous driving technology.

Here are some suggestions for articles:

1.     Chapter 2 is small and similar to chapter 1, would it be more appropriate to combine the two chapters?

2.     The innovations of this paper have not been fully elaborated. For the artificial intelligence-based LSTM localization method proposed in this paper, the principle and model structure, etc. are not described in detail.

3.     What is the role of Table 1 in the paper, it seems to be just an explanation of the document, which doesn't appear in the paper.

4.     4.1. Data Preprocessing should be section 4.2.

5.     The specific meaning of the symbols in each equation needs to be explained in section 5.1.

6.     It is recommended that a comparison with other methods be added to highlight the advantages of the methods in this paper.

Comments on the Quality of English Language

Moderate editing of English language required

Author Response

Foreword

Thank you for giving us the opportunity to submit a revised version of our paper "Deep Learning-Based Approach for Autonomous Vehicle Localization: Application and Experimental Analysis ". We appreciate the time and effort that to providing feedback on our paper and we are grateful for the insightful comments on and valuable improvements.

  1. Chapter 2 is small and similar to chapter 1, would it be more appropriate to combine the two chapters?

Thank you for this comment, indeed chapter 2 was too small, we combined with chapter 1 as suggested.

  1. The innovations of this paper have not been fully elaborated. For the artificial intelligence-based LSTM localization method proposed in this paper, the principle and model structure, etc. are not described in detail.

Thank you, we have extended the paper accordingly. We have extended the chapter 1.2 and we described similar approaches to ours. From the mentioned approaches [23] M. Brossard and S. Bonnabel, "Learning wheel odometry and IMU errors for localization," is the closest to ours. Tere are many small differences, but the main one is that instead of convolutional neural networks (CNN), our solution uses LSTM’s, a more advanced and recent type of NN.

  1. What is the role of Table 1 in the paper, it seems to be just an explanation of the document, which doesn't appear in the paper.

We have moved Table 1 into appendix, we agreed that this was not a vital part of the paper.

  1. 4.1. Data Preprocessing should be section 4.2.

Thank you, we have corrected it, please also note that because we merged section 1 and 2, the numbering changed.

  1. The specific meaning of the symbols in each equation needs to be explained in section 5.1.

We have added explanation of the symbols.

  1. It is recommended that a comparison with other methods be added to highlight the advantages of the methods in this paper.

We have extended the methodology part which explains what kind of benefits the proposed method have. Thank you for this remark too.

Reviewer 2 Report

Comments and Suggestions for Authors

Dear authors, 

Hereafter some comments from my side:

  • Overall, my general comment about the idea behind your work: I totally agree that model-based approaches – such as Extended Kalman Filter (EKF), Unscented Kalman Filter (UKF) and Particle Filter (PF) – are affected by noise in both the system and estimation models, as well as they necessitate an in-depth understanding of the non-linear model noise characteristics of the sensors and of the involved dynamics. Therefore, to properly tune these systems, it is needed a great investment of time and effort. However, it seems to me that you are shifting/moving the problem in some way, not completely solving it. In fact, Neural Networks (NNs) based systems have the big problem of generalization capability, that means the creation of an appropriate and adequate training dataset, with many different scenarios (conditions and situations) and enough number of samples. This is not always and necessarily easier to achieve.  
  • On page 4, figure 1, I think it will be helpful to add more information in the capture of the figure.
  • In some way, this is true also for the other figures, such as Fig. 2, and 4-8.
  • With reference to the point 1), on page 6, lines 241-250, when you write about pre-processing phase, this is true only for the training of NNs or in general?
  • In section 3.1, it would be nice to have more details about how the data collection has been performed, including the test-site/track you used.
  • In addition, in section 3.2, can you add more information about how the ground-truth data have been obtained?
  • In the same way, it is not totally clear how you got the estimated kinematic odometry model path. This can affect also how the conclusions are supported by the results.

I hope my comments can help to improve your worky.

Comments on the Quality of English Language

Just a minor check of text and formatting (e.g., on page 3, around line 120, "lover" instead "lower".

Author Response

Response to Reviewer’s comments

Foreword

Thank you for giving us the opportunity to submit a revised version of our paper "Deep Learning-Based Approach for Autonomous Vehicle Localization: Application and Experimental Analysis ". We appreciate the time and effort that to providing feedback on our paper and we are grateful for the insightful comments on and valuable improvements.

Overall, my general comment about the idea behind your work: I totally agree that model-based approaches – such as Extended Kalman Filter (EKF), Unscented Kalman Filter (UKF) and Particle Filter (PF) – are affected by noise in both the system and estimation models, as well as they necessitate an in-depth understanding of the non-linear model noise characteristics of the sensors and of the involved dynamics. Therefore, to properly tune these systems, it is needed a great investment of time and effort. However, it seems to me that you are shifting/moving the problem in some way, not completely solving it. In fact, Neural Networks (NNs) based systems have the big problem of generalization capability, that means the creation of an appropriate and adequate training dataset, with many different scenarios (conditions and situations) and enough number of samples. This is not always and necessarily easier to achieve. 

Thank you for this comment. We appreciate it and agree with your point of view. Creation of an appropriate and adequate training dataset, with many different scenarios is not easy and this aspect of the work was not emphasized enough. As you mentioned our approach does not completely solve the problem, it suggests only an alternative to Extended Kalman Filter (EKF), Unscented Kalman Filter (UKF) and Particle Filter (PF) and other well-known methods. We extended the paper to show that the proposed solution also has drawbacks compared to the well-established classical methods. Thus, a researcher who reads this paper can decide more confidently which approach to utilize.

On page 4, figure 1, I think it will be helpful to add more information in the capture of the figure.

In some way, this is true also for the other figures, such as Fig. 2, and 4-8.

Thank you, we have extended the description. Please note that the numbering of the figures have been changed.

With reference to the point 1), on page 6, lines 241-250, when you write about pre-processing phase, this is true only for the training of NNs or in general?

Thank you, this was not clearly emphasized, it only concerning the NNs. We extended the description.

In section 3.1, it would be nice to have more details about how the data collection has been performed, including the test-site/track you used.

Thank you now it is incuded.

In addition, in section 3.2, can you add more information about how the ground-truth data have been obtained?

We have described the two RTK GNSS equipment which were used at ground truth data acquisition. Thank you for the suggestions.

In the same way, it is not totally clear how you got the estimated kinematic odometry model path. This can affect also how the conclusions are supported by the results.

We used the single-track kinematic model, also known as the Kinematic Bicycle Model. This model can perform in real time, although, as mentioned it the paper, our current version cannot.

Just a minor check of text and formatting (e.g., on page 3, around line 120, "lover" instead "lower".

Thank you for spotting out this, it is corrected, and the grammar is checked once again.

Reviewer 3 Report

Comments and Suggestions for Authors

An AI-based long short-term memory (LSTM) localization approach, evaluates the performance. The topic is novel and has good potential for applying neural networks. In terms of the writing of the paper, please see my comments below:

- The abstract is not well written. There is too much background information but fails to introduce what has been done in the paper. Also, the last sentence is not complete. “with special focus on application” does not mention the example of the application.

- The main contributions of this paper are not elaborated in the introduction. Please consider how to describe the main contributions.

- The quality of the figures in the paper is good. However, please provide more information in the caption such as figure 2.

- The traditional Kalman filter-based localization approach has been widely used. Please consider discussing them in the paper as well. Some related works are: integrated inertial-LiDAR-based map matching localization for varying environments, secure cooperative localization for connected automated vehicles based on consensus, an automated driving systems data acquisition and analytics platform. In doing so, the interest of this paper will be much improved.

- When designing the network, please use some figures to show the structure of the neural networks. 

Author Response

Author's Notes to Reviewer 2

Foreword

Thank you for giving us the opportunity to submit a revised version of our paper "Deep Learning-Based Approach for Autonomous Vehicle Localization: Application and Experimental Analysis ". We appreciate the time and effort that to providing feedback on our paper and we are grateful for the insightful comments on and valuable improvements.

An AI-based long short-term memory (LSTM) localization approach, evaluates the performance. The topic is novel and has good potential for applying neural networks. In terms of the writing of the paper, please see my comments below:

Thank you for this comment.

- The abstract is not well written. There is too much background information but fails to introduce what has been done in the paper. Also, the last sentence is not complete. “with special focus on application” does not mention the example of the application.

Thank you for this remark, the last sentence was indeed not complete. Also we have been revised the abstract. We hope this version introduces our work better.

- The main contributions of this paper are not elaborated in the introduction. Please consider how to describe the main contributions.

Thank you, indeed this was not clearly written. We have extended the introduction section hopefully now it is much easier to understand it.

- The quality of the figures in the paper is good. However, please provide more information in the caption such as figure 2.

Indeed, the information e.g. in figure 2 was not clear enough. We revised the captions too.

- The traditional Kalman filter-based localization approach has been widely used. Please consider discussing them in the paper as well. Some related works are: integrated inertial-LiDAR-based map matching localization for varying environments, secure cooperative localization for connected automated vehicles based on consensus, an automated driving systems data acquisition and analytics platform. In doing so, the interest of this paper will be much improved.

Thank you, we have extended the first chapter (introduction) accordingly.  We have added integrated inertial-LiDAR-based map matching localization and related solutions too. You can find it int the extended introduction section.

- When designing the network, please use some figures to show the structure of the neural networks.

We have updated the network model architecture figure. About the structure of the network: LSTM’s hidden size was 128, the MLP head layer sizes are 64, 8, 3. The input (encoding) MLP size was 64. Dropouts are used with a value of 0.5 between MLP layers. The used sequence length was 5. We have also added this information to the paper. Thank you!

Round 2

Reviewer 1 Report

Comments and Suggestions for Authors

It is recommended that a comparison with other methods be added to highlight the advantages of the methods in this paper.

Comments on the Quality of English Language

 Moderate editing of English language required.

Author Response

Thank you for giving us the opportunity to submit a revised version of our paper "Deep Learning-Based Approach for Autonomous Vehicle Localization: Application and Experimental Analysis ". We appreciate the time and effort that to providing feedback on our paper and we are grateful for the insightful comments on and valuable improvements.

It is recommended that a comparison with other methods be added to highlight the advantages of the methods in this paper.

We have modified the Discussion section accourding to your comment. We have highlighted the advantages (also the drawbacks) of the methods in this paper.

Thank you for this remark.

Reviewer 3 Report

Comments and Suggestions for Authors

Appreciate the authors' response. However, the traditional Kalman filter-based localization approach has been widely used. Please consider discussing them in the paper as well. Some quite-relevant works is: integrated inertial-LiDAR-based map matching localization for varying environments. In doing so, the interest of this paper will be much improved.

Author Response

Thank you for giving us the opportunity to re-submit a newly revised version of our paper "Deep Learning-Based Approach for Autonomous Vehicle Localization: Application and Experimental Analysis ". We appreciate the time and effort that to providing feedback on our paper and we are grateful for the insightful comments on and valuable improvements regarding Kalman filter. 

Appreciate the authors' response. However, the traditional Kalman filter-based localization approach has been widely used. Please consider discussing them in the paper as well. Some quite-relevant works is: integrated inertial-LiDAR-based map matching localization for varying environments. In doing so, the interest of this paper will be much improved.

Thank you we have included an extended discussion traditional Kalman filter-based localization approaches. Now 8 new relevant wrok is dicussed such as integrated inertial-LiDAR-based map matching localization for varying environments and UAV Position Estimation and Collision Avoidance Using the Extended Kalman Filter.

Round 3

Reviewer 3 Report

Comments and Suggestions for Authors

The paper looks good now and I don't have further comments. 

Author Response

Thank you for this final review and giving us the opportunity to submit a revised version of our paper "Deep Learning-Based Approach for Autonomous Vehicle Localization: Application and Experimental Analysis ". We appreciate the time and effort that to providing feedback on our paper and we are grateful for the insightful comments on and valuable improvements.

Reply to the only comment in the review:

The paper looks good now and I don't have further comments. 

Thank you, we have slightly modified only the discussion section, as you mentioned that it can be improved. (Also we had a similar request from another reviwer)